# Methane Emissions Driven by Adding a Gradient of Ethanol as Carbon Source in Integrated Vertical-Flow Constructed Wetlands

**Xiaoling Liu** [1,†]**, Jingting Wang** [2,3,†]**, Xiaoying Fu** [3,†]**, Hongbing Luo** [2,4,5,*] 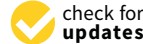**, Bruce C. Anderson** [6]**, Ke Zhang** [4,5]**, Mei Li** [7]**, Bo Huang** [8]**, Liangqian Fan** [4,5]**, Lijuan Yu** [8]**, Guozhu He** [8]**, Likou Zou** [9]**, Shuzhi Fu** [8]**, Limei Hu** [10] **and Mingshu Jiang** [11]

[1]  Department of Information Engineering, Sichuan Water Conservancy Vocational College, Yangma Town, Chengdu 611231, China; lxl.xc@163.com

[2]  College of Environmental Sciences, Sichuan Agricultural University, Chengdu 611130, China; abbywjt@126.com

[3]  State Key Laboratory of Hydraulics and Mountain River Engineering, Sichuan University, Chengdu 610065, China; xiaoyingfu62@163.com

[4]  College of Civil Engineering, Sichuan Agricultural University, Dujiangyan, Chengdu 611830, China; zhangke@sicau.edu.cn (K.Z.); flqjacky@163.com (L.F.)

[5]  Sichuan Higher Education Engineering Research Center for Disaster Prevention and Mitigation of Village Construction, Sichuan Agricultural University, Dujiangyan, Chengdu 611830, China

[6]  Department of Civil Engineering, Queen's University, Kingston, ON K7L 3N6, Canada; bruce.anderson@queensu.ca

[7]  School of Urban and Rural Construction, Chengdu University, Chengdu 610106, China; limeigm@126.com

[8]  Dujiangyan Campus, Sichuan Agricultural University, Chengdu 611830, China; huangbohuai@126.com (B.H.); lijuan_yu2003@126.com (L.Y.); heguozhu@sicau.edu.cn (G.H.); fushuzhi5555@gmail.com (S.F.)

[9]  Department of applied microbiology, Chengdu Campus, Sichuan Agricultural University, Chengdu 611130, China; zoulikou@sicau.edu.cn

[10]  Luzhou Environmental Protection Monitoring Center Station, Luzhou 646000, China; 15882466537@163.com

[11]  Sichuan Company of China Post Group, Chengdu 610006, China; jmsvicky@foxmail.com

[*]  Correspondence: hbluo@sicua.eud.cn or luohongbing66@163.com

[†]  These authors contributed equally to this work.

**Abstract:** This work aims to investigate the methane emissions from integrated vertical-flow constructed wetlands (IVCWs) when ethanol is added as an external carbon source. In this study, a gradient of ethanol (0, 2, 4, 8, 16 and 32 mmol/L) was added as the carbon source in an IVCW planted with *Cyperus alternifolius* L. The results showed that the methane emission flux at an ethanol concentration of 32 mmol/L was 32.34 g $CH_4$ $m^{-2}$ $day^{-1}$ less than that of the control experiment (0 mmol/L) and that the methane emission flux at an ethanol concentration of 16 mmol/L was 5.53 g $CH_4$ $m^{-2}$ $day^{-1}$ less than that at 0 mmol/L. In addition, variations in the water quality driven by the different ethanol concentrations were found, with a redox potential range of $-64$ mV to $+30$ mV, a pH range of 6.6–6.9, a chemical oxygen demand (COD) removal rate range of 41% to 78%, and an ammonia nitrogen removal rate range of 59% to 82% after the ethanol addition. With the average $CH_4$-C/TOC (%) value of 35% driven by ethanol, it will be beneficial to understand that $CH_4$-C/TOC can be considered an ecological indicator of anthropogenic methanogenesis from treatment wetlands when driven by carbon sources or carbon loading. It can be concluded that adding ethanol as an external carbon source can not only meet the water quality demand of the IVCW treatment system but also stimulate and increase the average $CH_4$ emissions from IVCWs by 23% compared with the control experiment. This finding indicates that an external carbon source can

stimulate more $CH_4$ emissions from IVCWs and shows the importance of carbon sources during sewage treatment processes when considering greenhouse emissions from treated wetlands.

**Keywords:** methane emission; integrated vertical-flow constructed wetland; ethanol; carbon source

## 1. Introduction

Constructed wetlands (CWs) are an engineered form of wetland that utilizes the natural environment—substrates, microorganisms, and plants—as a form of water treatment, mimicking a naturally occurring wetland [1] and are designed to take advantage of naturally occurring processes involving wetland plants, soil matrices and microorganisms for the wastewater treatment [2]. Constructed wetlands are divided into several specific types according to their construction and operation conditions. The integrated vertical-flow constructed wetland (IVCW) is a special wastewater treatment system that has a nice wastewater treatment effect and functions of landscape design and viewing. IVCW consists of a down-flow tank and an up-flow tank. This U-shaped flow structure gives rise to the metagenic "aerobic-anoxic-anaerobic-anoxic-aerobic" multifunctional layers [3]. The structure of the influent tank allows it to function as an aerobiotic zone, whereas the effluent tank is an anaerobic zone [4]. However, these structures and operations can lead to the accumulation of methanogens as a precondition of methane emissions [5].

Methane makes a significant contribution to the global climate change [6], and the continuous methane emissions may cause further warming and changes in many components of the climate system [7,8]. Therefore, research on methane emissions and their required controls in wastewater treatment systems is an important issue [9]. Climate-driven fluctuations of the $CH_4$ emissions from natural wetlands (177 to $284 \times 10^{12}$ g $(CH_4)$ yr$^{-1}$), and anthropogenic emissions account for 50 to 65% of total emissions of the global methane [10]. The methane emission contribution from all wetlands (natural and constructed wetlands) accounts for 70% of the total methane sources from nature [10,11], and constructed wetlands might play an indispensable part in the methane production and global warming potential [12]. Methane emissions from constructed wetlands are produced by microorganisms such as methanogens and methane-oxidizing microorganisms [13]. In addition, the methane production can be controlled by the functioning of microbiota. Microbiota affected by environmental factors such as the types of carbon source or the mass ratio of the total carbon to total nitrogen (the C/N ratio), temperature and presence of aeration [14]. Some researchers have studied the effects of aeration and oxidation-reduction potential on methane emissions [15,16], and few have studied the impact of carbon sources on methane emissions.

The carbon source is a limiting factor during the degradation of organic matter in constructed wetlands [17], including IVCWs [18]. The carbon sources in constructed wetlands include the carbon contained in wastewater, internal carbon sources and external carbon sources. Internal carbon sources include microbes or plant root secretions of organic matter, plant litter decomposition of organic matter and a matrix of sedimentary organic matter. In IVCW, the internal carbon includes organic matter from microbes, substrates, plant root secretion, and plant litter decomposition. External carbon sources consist of sugars (e.g., glucose, sucrose, fructose) and the soluble carbon (e.g., methanol, ethanol, acetic acid) from easily biodegradable carbon sources, natural plant materials (e.g., plant straw, agricultural waste, plant leaves) and natural organic matter (e.g., paper, cotton, and rice husk) [19]. The main effect of carbon sources on microbial growth is to provide the energy for cell life activities. Microbes consumption or carbon sources are necessary for microbial metabolism [20,21]. When the carbon content is the same, the carbon removal efficiency varies in the reaction system with the different kinds of carbon sources [22], and the authors found the rate constant of substrate degradation in the reaction system when using sucrose, glucose, sodium acetate and propionic acid as the carbon source was 0.117, 0.123, 0.055, and 0.056 mL mg$^{-1}$ d$^{-1}$, respectively. The optimum glucose dosage was determined to

be 1.5 g for the 60 L d$^{-1}$ IVCW treatment system [23]. It was found that the optimum dosage of carbon source or C/N ratio could be considered as two important factors to assess the carbon source effects. The optimum C/N ratio when using glucose as a carbon source was 7:1–6:1 ($C_6H_{12}O_6$:$NO_3^-$-N) [24]. The carbon source is oxidized during the process of aerobic oxidation, and the internal carbon from the rhizosphere is consequently insufficient for denitrification. In addition, adding external carbon sources is typically done to adjust the C/N ratio in constructed wetlands [25,26]. The wastewater must contain a sufficient amount of a carbon source including matters or substrates, to ensure bacterial metabolism [27]. Extra carbon sources need to be added to address an imbalance in the C/N ratio when the influent C/N ratio is below 3.4 [28]. The C/N ratios of chemical wastewater, aquaculture wastewater, landfill leachate, surface runoff and some municipal wastewater are all lower than this value [29]. Therefore, it is necessary to add an extra carbon source to balance the nutrition ratio of the biochemical system in constructed wetlands when necessary. Currently, external carbon sources have attracted attention for increasing the COD (Chemical Oxygen Demand)/$NO_3$-N ratio of the influent wastewater to enhance the total nitrogen removal efficiency [30–34]. These available organic carbon sources enhance the nitrogen removal efficiency in the wastewater treatment [35,36]. Microorganisms tend to metabolize simple and smaller molecular carbon sources.

Although methanotrophs and ammonium oxidizers play an important role in the global C and N cycles, their ecological interactions are not well understood [37]. According to previous studies, increased carbon loading has been found to enhance the $CH_4$ emissions [38]. A higher inflow loading of both total organic carbon (TOC) and total nitrogen (TN) always increases the respective $CH_4$ and $N_2O$ fluxes [39,40]. This is also demonstrated by the significant correlation between the inflow TOC and TN values and the corresponding $CH_4$ and $N_2O$ fluxes [41]. The non-linear character of the $CH_4$-C emissions vs. the $TOC_{in}$ relationship in the horizontal subsurface flow (HSSF) and vertical subsurface (VSSF) CWs may indicate some limit level for the $CH_4$ release; i.e., for a certain $TOC_{in}$ loading, methanogenesis may be inhibited by an increased concentration of ammonia or by an accumulation of volatile fatty acids inhibiting the rate limiting step of methanogenesis or the hydrolysis of the organic matter [41,42]. Thus, the $CH_4$-C, TOC, inorganic carbon (IC) and total carbon (TC), as indicators of greenhouse gas emissions, are significantly related to the carbon budget and treatment situations of the sewage and industrial wastewater pollution by the CW technology. The $CH_4$ emissions significantly influenced the total carbon budget from a *Phragmites australis* wetland in the Zhangye oasis-desert area [43]. Additionally, despite some uncertainties, the soil organic carbon (SOC) is a good proxy for predicting both annual $CH_4$ and $N_2O$ emissions, which were described well by decaying power regressions as a function of the SOC content in the range of 6–23% C [44]. Currently, the wastewater with a high organic concentration is much more common in wetlands, constructed wetlands and sewage treatment systems [45–47]. Large amounts of methane are produced during the process of anaerobically treating wastewater with high concentrations of organic carbon, mainly from the sewage sludge stabilization in municipal and industrial Wastewater Treatment Plants (WWTPs) [48,49]. Therefore, it is important to understand how the carbon budget, including external and internal carbon sources, impacts the $CH_4$ emissions from treatment wetlands.

Methanol and ethanol are ideal carbon sources [50], and sucrose, methanol, and ethanol have been used as carbon sources to treat the contaminated groundwater. The results showed that when sucrose was used as a carbon source, nitrite was produced. The effect of using methanol and ethanol as carbon sources on the dissolved oxygen is much less pronounced than that of sucrose. The biological denitrification, on the other hand, uses simple compounds as carbon sources. However, methanol is not widely used because of its toxicity, high cost and inconvenient transportation. According to previous findings, the utilization rate of ethanol as a carbon source is three times that of methanol in constructed wetlands [51]. Ethanol is very effective in removing nitrate in water [51], and the ethanol production is economically viable from sugar crops [52]. The raw material of ethanol comes primarily from corn, cassava, molasses, and sugar cane during agricultural and industrial processes. Moreover, there is considerable research on methods to produce ethanol at a low cost. For instance, the ethanol production

from conventionally farmed seaweed could cost less than the conventional ethanol and be produced on a scale comparable to 1% of the global gasoline production [53]. According to Magyar et al. [54], sugar-rich food waste is a sustainable feedstock that can be converted into ethanol without an expensive thermochemical pretreatment. Because the costs of the ethanol production are decreasing, ethanol can be considered a promising carbon source to add to water for pollutant degradation and methane emission control in constructed wetlands. Ethanol as a carbon source has been applied in many situations. Ethanol can be used to prepare nanomaterials, which are difficult to form from the amorphous carbon, because alcohol is not only non-toxic but also easy to store and transport [55].

Some researchers have explored the dissolved organic carbon in natural wetlands [56], but there is still no knowledge of the methane emissions from constructed wetlands when adding ethanol as an external carbon source.

IVCWs are effective for wastewater purification, but adding an extra carbon source may affect the methane emissions from the IVCW. To evaluate ethanol as a carbon source and its effect on the methane emissions in constructed wetlands, the objectives of this study were (1) to analyze the rate of the methane emissions from six different concentrations of ethanol added into IVCW; (2) to investigate the effects of ethanol in the methane flux and the carbon usage rate and assess the indication of $CH_4$-C/TOC, and at the same time, (3) to explore the dynamic kinetic features of the methane emissions in an IVCW driven by different ethanol concentrations, which may have an impact on the methane emissions.

## 2. Material and Methods

### 2.1. Study Area

The experiment described in this paper was conducted at the College of Civil Engineering, Sichuan Agricultural University, Dujiangyan City, Sichuan Province. Dujiangyan City (longitude: 103°25′–103°47′ E, latitude: 30°44′–31°22′ N) is part of the administrative Chengdu City, Sichuan province. Dujiangyan City has a humid subtropical climate and four distinct seasons, with temperatures ranging from −5 °C to 32 °C. The average temperature was lowest at 4.6 °C in January and highest at 24.7 °C in July, with an annual average temperature of 15.2 °C.

### 2.2. Integrated Vertical-Flow Constructed Wetland

The experimental design consisted of an integrated vertical-flow constructed wetland (1000 mm length × 1000 mm width × 700 mm depth), with influent and effluent tanks made of plexiglass with 10 mm for the tank wall and tank bottom (Figure 1). The IVCW has a slope of 10% on the tank bottom. The substrate height of layer 1, layer 2, layer 3, layer 4 and layer 5 in IVCW was 150, 100, 150, 50, 150 mm, respectively. The substrates in the IVCW included five layers from top to bottom (Luo et al., 2015): The first layer (surface layer) was fine sand with a particle size of less than 1 mm (the mean permeability was 10 m d$^{-1}$); the second layer was coarse sand with a particle size of 1–2 mm (the mean permeability was 45 m d$^{-1}$); the third layer was gravel with a particle size of 2–10 mm (the mean permeability was 250 m d$^{-1}$); the fourth layer was gravel with a particle size of 10–30 mm (the mean permeability was 600 m d$^{-1}$); and the fifth layer (bottom layer) was a layer of pebbles with a particle size of 30–60 mm (the mean permeability was 1100 m d$^{-1}$). The effect volume filling water in the IVCW was 125 L. Perforated pipes (PVC tubes with inner diameter of 10 mm) were used for water distribution from the influent tank to the effluent tank. Table 1 details the water sampling points within the IVCW. These points were divided into five horizontal layers (Plane I, Plane II, Plane III, Plane IV, and Plane V) within the influent tank (named B), and the effluent tank (named D). *Cyperus alternifolius* L. was used as the wetland plant due to its strong ability to absorb pollutants and grow throughout the year. *Cyperus alternifolius* L. was planted as the first layer, and 16 plantlets were planted in the influent tank and effluent tank separately.

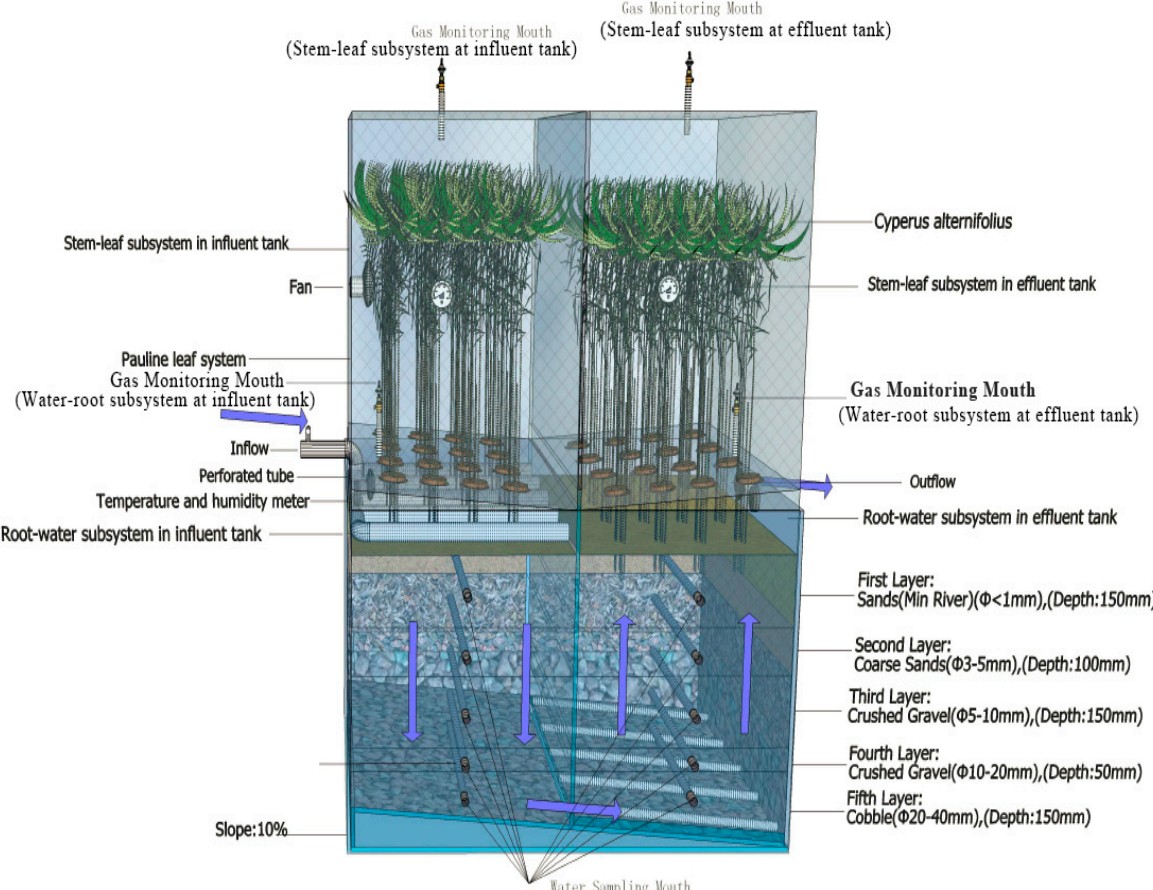

**Figure 1.** Distribution of substrate layers, sampling pipes and greenhouse gas collecting device of integrated vertical-flow constructed wetlands (IVCW). (ø is the diameters of pipe).

**Table 1.** Distribution of internal sampling points in the influent and effluent tank.

| IVCW | Group B (Influent Tank) | Group D (Effluent Tank) | The Distance from the Sampling Spots to Bottom of the Tank | Position |
|---|---|---|---|---|
| Plane I | I-B3 | I-D3 | 525 mm | The middle position of first layer |
| Plane II | II-B3 | II-D3 | 400 mm | The middle position of second layer |
| Plane III | III-B3 | III-D3 | 300 mm | The middle position of third layer |
| Plane IV | IV-B3 | IV-D3 | 200 mm | The middle position of fourth layer |
| Plane V | V-B3 | V-D3 | 75 mm | The middle position of fifth layer |

## 2.3. Experimental Design and Operation

The methane and carbon dioxide collection system was divided into four parts along the plant body: Stem-leaf subsystem and root-water subsystem in both influent and effluent tanks. Every subsystem was isolated and sealed with plastic wrap and glued to ensure that the gas samples came only from those subsystems [57]. The methane collection system was placed on the top of the IVCW because methane is less dense than air. The influent and effluent tanks were installed with thermometers, hygrometers and fans in the methane collection subsystems (Figure 1). The methane collection system was opened, and the fans were operated to remove the residual gas after the gas sample was collected each time. The influent velocity of ethanol solution is 10.44 L h$^{-1}$, and the influent is stopped until the water depth is six centimeters deep.

Each experiment included the following steps: (1) To clean and maintain the normal status of the IVCW for preparing the water treatment, the IVCW was cleaned with tap water. The water was released in the constructed wetland first, and flow was maintained in and out for an hour to ensure that no residual ethanol would affect the next experiment. (2) A concentration of ethanol solution was

added to the IVCW until it was completely filled. (3) The seals of the gas collection system were ensured to be closed. (4) To set the experimental starting time, the authors referred to the microbial metabolism of the carbon source in the constructed wetland. Generally, there are small changes between 0 and 16 h in the metabolic ability; this ability gradually strengthens from 16 to 96 h and then stabilizes after 96 h (Deng et al., 2011). Based on these observations, the experiment started at 18:30 on the first day. Sampling was done at 0, 6, 12, 18, 24, 32, 40, 48, 60, 72, 96 and 120 h. (5) To ensure the comparability of the experiments, the corresponding water samples at the positions of group B and group D in the IVCW were taken when the gas samples were analyzed for methane gas, with the water samples analyzed for COD, TC, TOC, IC, TN, $NH_3$-N, pH, Eh, and temperature. (6) To prevent the residual gases from interfering with the next concentration experiment, at the end of each experiment, gas discharge switches were turned off and the plastic gas collection system was opened for half an hour to emit the concentrated methane and carbon dioxide. (7) To remove the remaining methane and $CO_2$, residual air was released by fully opening all plastic films of the gas collection subsystems and turning on the gas monitoring mouths to conserve the planted *Cyperus alternifolius L.* by exchanging air.

The duration of each experiment was 120 h (five days). However, methanogens have a metabolic need to consume large amounts of a carbon source. The glucose consumption by methanogens from 1000 mg·$L^{-1}$ to 100 mg·$L^{-1}$ takes 15 h [22]. Plants also need a sufficient carbon source to promote growth. Therefore, the operation steps in each experiment were repeated with the addition of 0 mmol/L, 2 mmol/L, 4 mmol/L, 8 mmol/L, 16 mmol/L and 32 mmol/L (92, 184, 368, 736 and 1472 mg·$L^{-1}$) ethanol as the carbon source in turn. Tap water (0 mmol/L ethanol) was added to the constructed wetland as the control experiment.

### 2.4. Carbon Utilization Analysis

According to the principle of mass conservation, assessing the carbon balance of a constructed wetland is necessary for sustainable development and to decrease the greenhouse effect. The following equations were used to assess the balance of the extra carbon source in the IVCW. The carbon balance model used in this paper is a simple representation of the carbon cycle in the ecosystem of the constructed wetlands [58,59].

$$C_{input} = C_{output} + C_{residual} \tag{1}$$

$$C_{residual} = C_{water} + C_{plant} + C_{substrate} \tag{2}$$

where $C_{input}$ is the total carbon (mg) from the influent tank and the effluent tank when adding different ethanol solutions as the carbon source. $C_{output}$ is the carbon emission ($CH_4$ and $CO_2$, mg) from the IVCW after adding different types of ethanol as the carbon source. $C_{water}$, $C_{plant}$, and $C_{substrate}$ are the residual carbon mass (mg) from the water, plant and substrate in the IVCW system, respectively.

### 2.5. Analytical Methods

The methane fluxes ($CH_4$ fluxes) and carbon dioxide concentrations ($CO_2$) were determined by the infrared spectrophotometry using a portable soil gas flux measurement system (Portable WS-LI820, WEST Ltd., Firenze, Italy). The operating steps were as follows: (1) Connect the gas chamber to the gas collection system, and seal it with a hose. (2) Connect the portable computer and the soil gas flux system via the bluetooth. (3) After preheating, start the air-opening switch, and click the start icon button to begin the determination. (4) Test each sample for 300 s. The TC, TOC and TN were determined by a catalytic oxidation method (multi N/C 2100, Analytikjena Ltd., Jena, Germany) and the inorganic carbon (IC) by subtracting TOC from TC. The redox potential (Eh) and pH were measured by acidity meters (PB-10, Sartorius Corporate, Otto-Brenner-Straße, Goettingen City, Germany), and COD was measured by the potassium dichromate titrimetric method (Stanislaw and Jenkinson 1973). $NH_3$-N was determined by the indophenol blue method (Stanislaw and Jenkinson 1973). The sum of the $CH_4$

emissions was converted to $CO_2$ equivalent emissions using 100-year global warming potential (GWP) coefficients, i.e., 23 for $CH_4$ [60].

## 2.6. Statistical Analysis

The influence of methane emission flux parameters (the sampling system and ethanol concentration) was determined by tests of between-subjects effects, each of which was conducted through a mixed-design ANOVA to analyze its relationship to the control experiment. The multivariate analysis of the variance and principal component analysis was analyzed between the methane emissions and other factors (ethanol concentration, temperature, Eh and pH). We used the free software package, R Foundation for Statistical Computing (version 3.3.1) to perform the statistical analysis. The dynamic kinetic model of the methane emissions driven by the ethanol addition was determined by the 1stOpt (First Optimization) free (Ver.1.5.0). For all statistical tests, the significance was considered at $p$ values below 0.05.

## 3. Results

### 3.1. Methane Emissions and Effects on Water Quality Driven by Ethanol Addition in the IVCW

The methane emissions including the control experiment (0 mmol/L of ethanol) are shown in Table 2. The methane emissions from the IVCW at 2 mmol/L, 4 mmol/L and 8 mmol/L of the ethanol concentrations were all higher than that of the 0 mmol/L (control experiment) emissions. The $CH_4$ emissions from the external carbon source (ethanol) experiment of 2 mmol/L, 4 mmol/L, 8 mmol/L, 16 mmol/L and 32 mmol/L in Table 2 increased by 42%, 42%, 29%, 33%, and −35%, respectively, compared with the control experiment (0 mmol/L). The multivariate analysis of the variance and principal component analysis was analyzed between the methane emissions and other factors (ethanol concentration, temperature, Eh and pH), which the use of 4 mmol/L ($p$ value = 0), 8 mmol/L ($p$ value = 0.001) and 32 mmol/L ($p$ value = 0.002) of ethanol as a carbon source showed a significant difference from the use of 0 mmol/L (control experiment).

**Table 2.** Methane emission flux of each ethanol concentration in the IVCW.

| Methane Fluxes (g $CH_4$ $m^{-2}$ $day^{-1}$) | Ethanol Concentrations | | | | | |
|---|---|---|---|---|---|---|
| | 0 mmol/L (Blank Experiment) | 2 mmol/L | 4 mmol/L | 8 mmol/L | 16 mmol/L | 32 mmol/L |
| Total methane flux in IVCW | 101.5 ± 0.07 | 146.49 ± 0.08 | 146.19 ± 0.08 | 133.44 ± 0.09 | 106.88 ± 0.04 | 69.19 ± 0.05 |
| Stem-leaf subsystem in influent tank | 25.07 ± 0.05 | 36.94 ± 0.07 | 31.47 ± 0.06 | 36.83 ± 0.08 | 26.32 ± 0.04 | 14.69 ± 0.04 |
| Root-water subsystem in influent tank | 22.96 ± 0.09 | 36.14 ± 0.08 | 39.55 ± 0.09 | 30.91 ± 0.06 | 25.84 ± 0.04 | 17.81 ± 0.05 |
| Stem-leaf subsystem in effluent tank | 26.70 ± 0.08 | 45.42 ± 0.10 | 35.74 ± 0.07 | 29.02 ± 0.06 | 25.78 ± 0.04 | 14.976 ± 0.06 |
| Root-water subsystem in effluent tank | 28.72 ± 0.09 | 27.95 ± 0.05 | 39.44 ± 0.08 | 36.66 ± 0.08 | 28.95 ± 0.06 | 20.72 ± 0.06 |
| Added methane flux of control experiment, compared with blank experiment | - | 43.04 ± 0.25 (42%) | 42.74 ± 0.31 (42%) | 29.98 ± 0.37 (29%) | 3.42 ± 0.24 (33%) | −35.26 ± 0.27 (−35%) |

Figure 2 illustrates the methane emissions from the IVCW at six concentrations of ethanol collected by four gas-sampling subsystems as a function of sampling time. There was an increasing tendency of the methane emissions from the IVCW when 2 mmol/L and 4 mmol/L of ethanol were added. Furthermore, for the 2 mmol/L of ethanol addition, the methane emission from the IVCW reached the lowest flux (6.94 g $CH_4$ $m^{-2}$ $day^{-1}$) at the sampling time of 24 h, and the methane emission had a stable increase for the remainder of the time. The highest flux value at 2 mmol/L of ethanol addition was 18.22 g $CH_4$ $m^{-2}$ $day^{-1}$ at 120 h. The minimum methane flux (Figure 2c) was observed for the 4 mmol/L addition (3.87 g $CH_4$ $m^{-2}$ $day^{-1}$), and the methane emissions in the other monitoring periods were in the range of 9.60–19.20 g $CH_4$ $m^{-2}$ $day^{-1}$. The 8 mmol/L dose (Figure 2d) showed a decreasing trend throughout the test, and the lowest methane flux of this concentration (4.48 g $CH_4$ $m^{-2}$ $day^{-1}$) from the IVCW in the experimental period occurred at 72 h. The methane emissions remained at the stable flux at the 16 mmol/L (Figure 2e) ethanol dose. Moreover, the methane flux

from the IVCW at 16 mmol/L (Figure 2e) slightly increased, while the 32 mmol/L dose flux (Figure 2f) was lower than the control experiment (0 mmol/L). For the 32 mmol/L dose (Figure 2f), no values were less than 8.58 g $CH_4$ $m^{-2}$ $day^{-1}$. The average value of the root-water subsystem was higher than that of the stem-leaf system.

Figure 3 shows the changes in the physicochemical parameters of the temperature, pH and Eh in the IVCW at different concentrations of ethanol. The range of temperature change in the natural condition was very narrow (Figure 3c). Moreover, the minimum emissions of the methane occurred when the temperature was the lowest, and 32 mmol/L of ethanol was used as the carbon source (8.76 °C at 40 and 72 h). The pH (Figure 3a) values ranged from 6.8 to 7.0 at the ethanol concentrations of 2 mmol/L, 4 mmol/L, 8 mmol/L and 16 mmol/L. When 32 mmol/L of ethanol was added, the pH ranged from 6.5 to 6.8. The pH value at the 16 mmol/L of ethanol dose was highest, whereas the 32 mmol/L of ethanol dose had the lowest value at each corresponding sampling point location (Figure 3a). The pH values had a consistent tendency, except at the 4 mmol/L of ethanol dose, where the pH slowly decreased to the lowest sample point (II-D-3) and then increased again. The pH in the IVCW at 32 mmol/L sharply and quickly decreased to 6.6, rose slightly and was less than the corresponding points of the control experiment. Changes in the water system pH can influence the environment or the growth of methanogens, in turn affecting the methane emissions.

The redox potentials of the 2 mmol/L and 4 mmol/L doses were in the range of −20 mV to 0 mV (Figure 3b), whereas those of the 8 mmol/L and 16 mmol/L of ethanol doses were both in the range of −40 mV to −20 mV. These values are very high compared to those of the typical anaerobic systems generating methane (−300 mV), which indicates that this is another important factor influencing the methane generation. The average redox potentials of the 0 mmol/L and 32 mmol/L doses were higher than the other doses, and the methane fluxes were lower. Figure 3b also shows that the redox potentials in the IVCW for the control experiment were in the range of −7 mV–5 mV, whereas the redox potentials in the IVCW for the other five ethanol concentrations showed lower values than the control experiment after the addition of ethanol. Moreover, the redox potential ranges for the 2–16 mmol/L of ethanol doses were very similar, with average redox values of −24 mV, −23 mV and −26 mV, respectively. In addition, the average redox value for the 8 mmol/L dose was −33 mV, and the range was wider (−32 mV to −27 mV) than those of the 2 mmol/L, 4 mmol/L and 16 mmol/L doses. The methane emissions thus have a correlation with the redox potential and pH, but the correlation was not significant in this study ($p = −0.31$, $p = 0.18$).

Figure 4 summarizes the statistical values for the removal rate of the chemical oxygen demand and ammonia nitrogen in the IVCW under the six ethanol doses. The min, median, mean and max value of the removal rate of COD was 41%, 48.3%, 51% and 78%, respectively, in Figure 4a. The min, median, mean and max value of the removal rate of ammonia nitrogen was 10.2%, 58%, 57%, and 93%, respectively, in Figure 4b. The highest removal rate of COD (16 mmol/L dose) was 78% at 18 h, which the average pH value of 16 mmol/L of ethanol was highest and possibly accelerated the COD degradation. The peak average value of the ammonia nitrogen removal was 61% when 16 mmol/L of ethanol was added. The removal rate of ammonia nitrogen increased after adding 2 mmol/L, 4 mmol/L, 8 mmol/L and 16 mmol/L of ethanol compared with 0 mmol/L of ethanol, which shows that a carbon source less 16 mmol/L of ethanol could effectively stimulate biodegradation in the constructed wetland (CW) in this study.

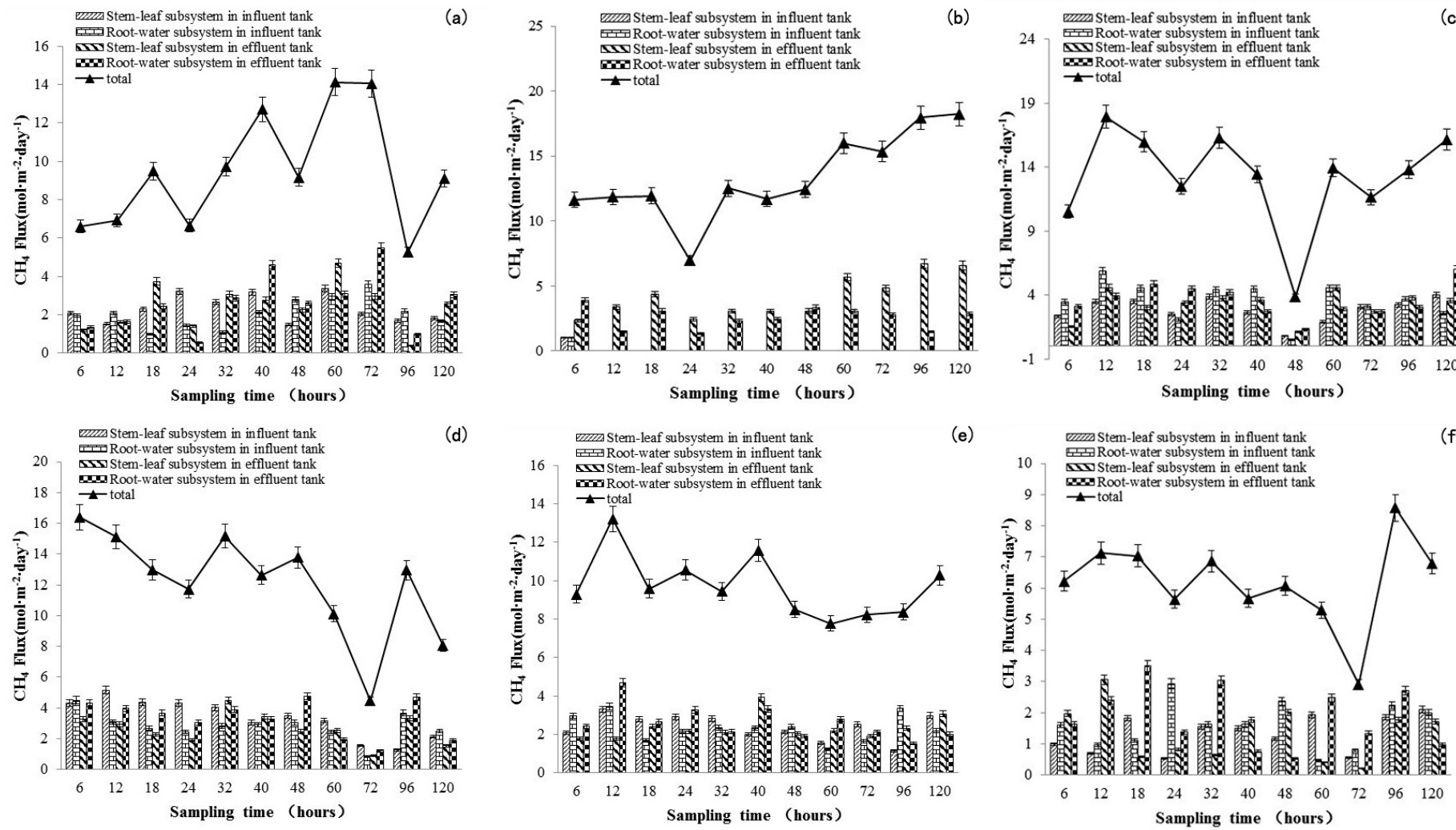

**Figure 2.** The methane emissions fluxes from the IVCW. (**a**) 0 mmol/L ethanol added; (**b**) 2 mmol/L ethanol added; (**c**) 4 mmol/L ethanol added; (**d**) 8 mmol/L ethanol added; (**e**) 16 mmol/L ethanol added; (**f**) 32mmol/L ethanol added.

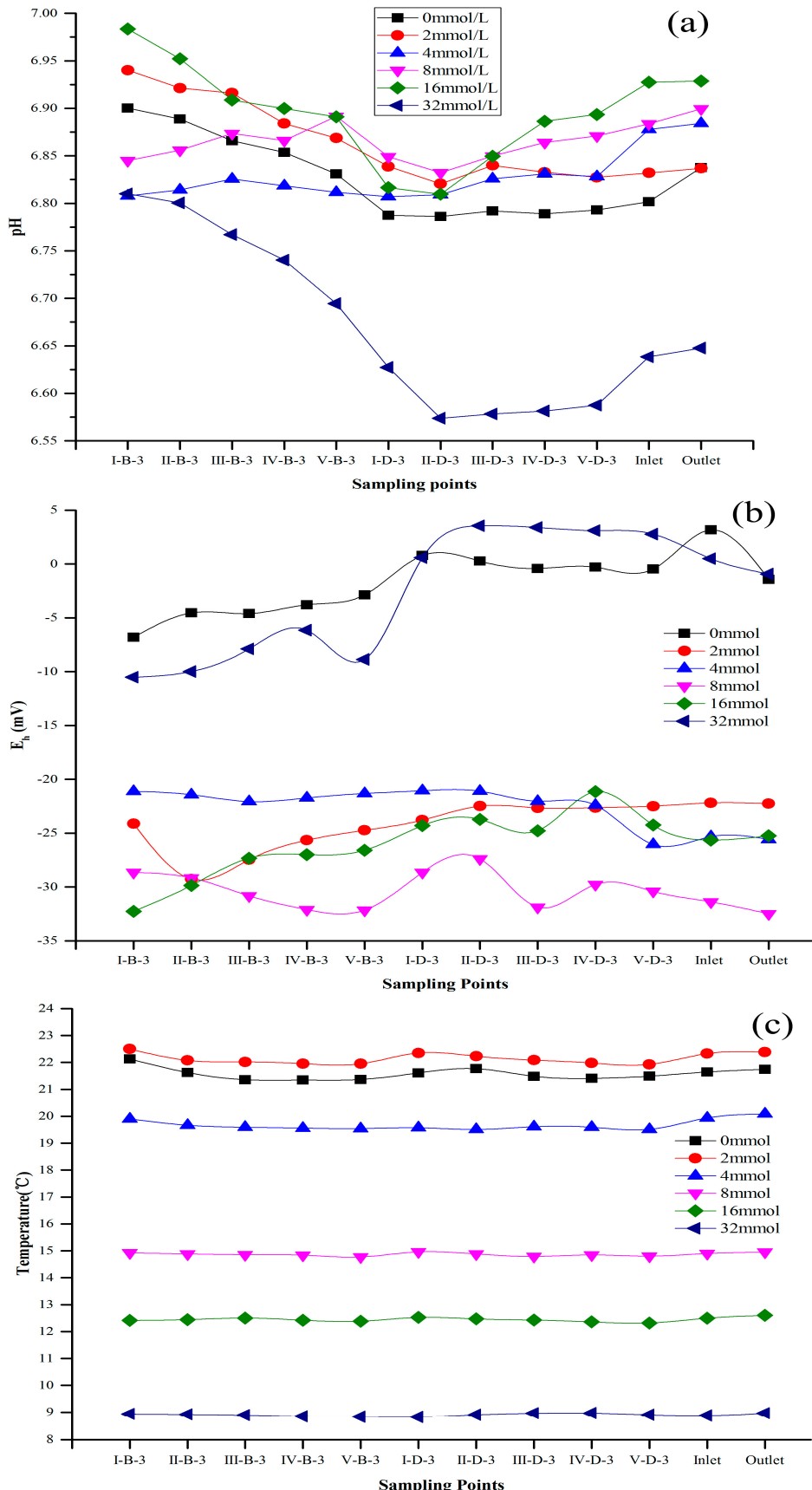

**Figure 3.** The parameters of sampling points in the IVCW (**a**) pH, (**b**) Eh, (**c**) temperature.

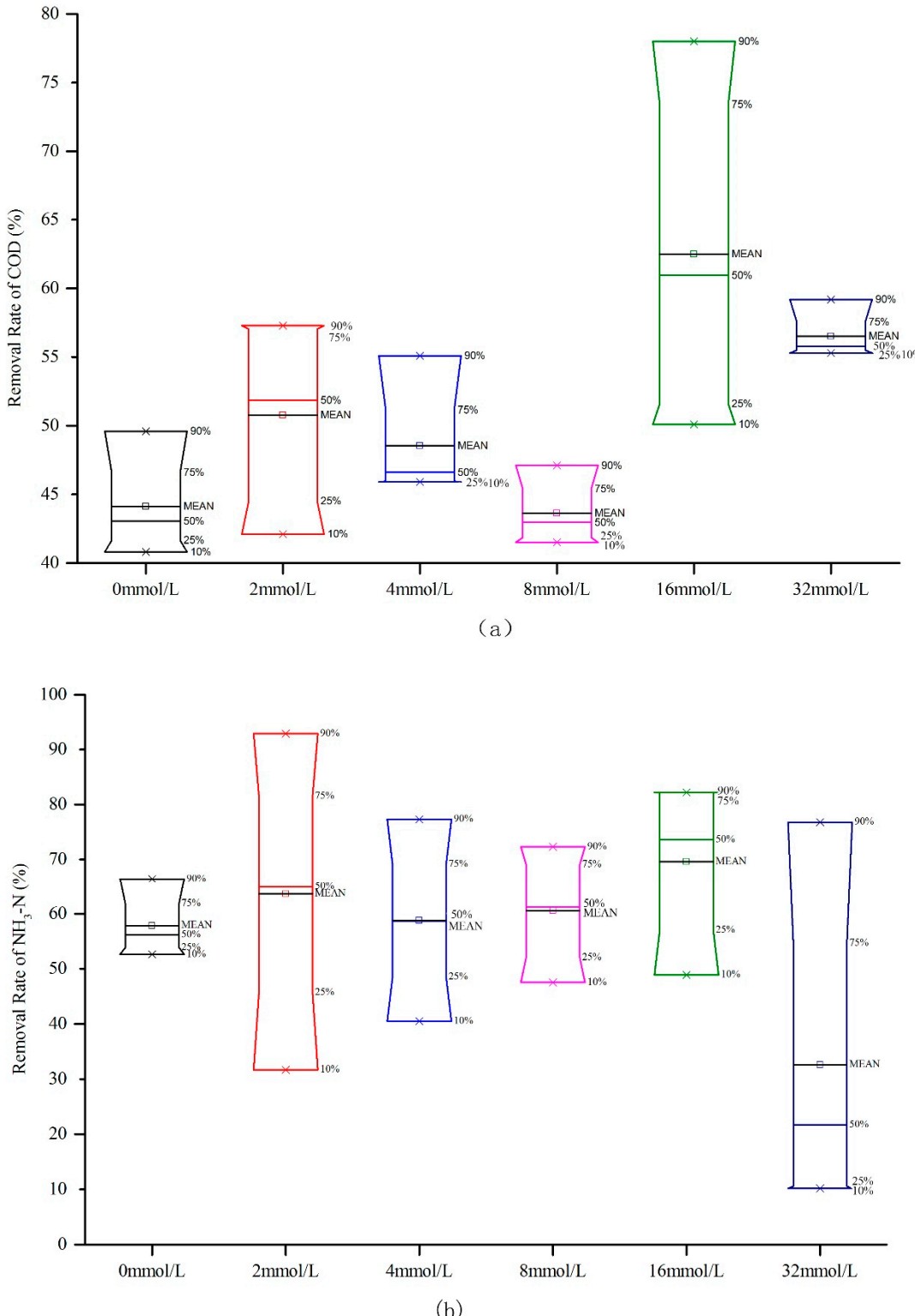

**Figure 4.** Removal rate of the chemical oxygen demand (COD) (**a**) and NH$_3$-N (**b**) with a gradient ethanol concentrations.

The regression analysis (Equation (3)) and principal component analysis (Equation (4)) was conducted considering the methane emissions and other factors (ethanol concentration, temperature, Eh and pH). Equation (3) shows a negative correlation between the methane emission and ethanol

concentration. Equation (4) shows that the ethanol concentration and temperature ($p < 0.01$) made major contributions to the methane emissions from each of the factors.

$$Y_{methane} = -2.68_{ethanol} + 153.72, \mathrm{R}^2 = 0.994 \qquad (3)$$

$$Y_{methane} = -0.992_{ethanol} + 0.988_{temp} + 0.107_{Eh} + 0.149_{pH}, \mathrm{R}^2 = 0.91 \qquad (4)$$

### 3.2. Carbon utilization of Different Ethanol Concentrations in the IVCW

Figure 5 displays the different carbon mass usage percentage at six ethanol addition concentrations compared with the sampling time. The average baseline carbon utilization (0 mmol/L dose) was 1% and remained stable. The average usage rate at 2 mmol/L of ethanol was 38%, and at 4 mmol/L of ethanol, it was 63%. The carbon utilization at these ethanol doses was variable, with initial increases followed by decreases, and both reached their highest values (65% and 82%, respectively) at 72 h. The average usage rate was 80% at 8 mmol/L of ethanol and 86% at 32 mmol/L of ethanol. Moreover, the highest average usage rate (88%) was observed when 16 mmol/L of ethanol was added. Interestingly, at the 16 mmol/L dose, the average usage rate from 6 to 48 h (84%) was less than that for the 60–120 h period (91%). Therefore, it was concluded that the *Cyperus alternifolius L.* and microorganisms in the IVCW have high utilization rates for the high concentration of ethanol (8–32 mmol/L).

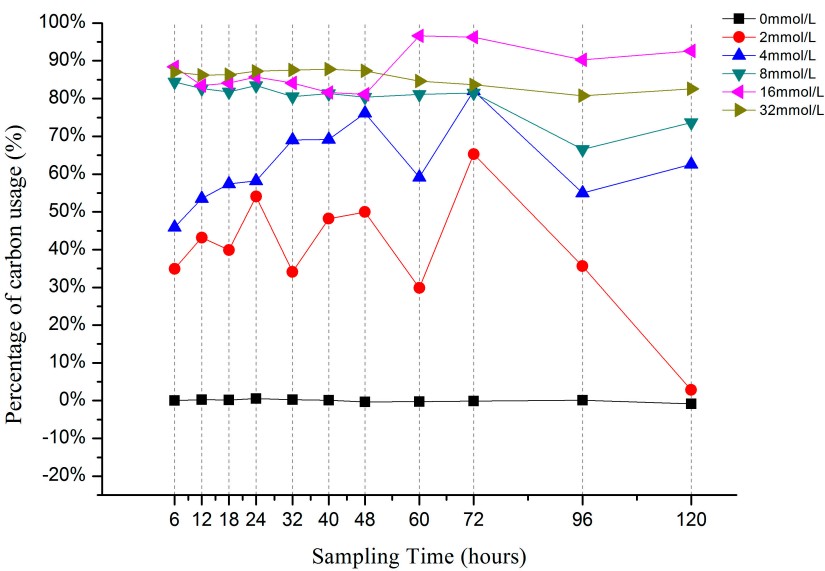

**Figure 5.** Variation in carbon mass usage percentage.

The outflow of total carbon and total nitrogen gradually reduced with the consumption of plants and microbes compared with the inflow. Table 3 shows the C/N ratio in the IVCW as driven by a gradient of ethanol. The inflow of the C/N ratio (15.6–193.8) increased by an increase in the concentration (0–32 mmol/L) of ethanol ($\mathrm{R}^2 = 0.92$). The different methane emissions under the six concentrations of ethanol may be attributed to the inflow C/N ratio and the possible effect of the 32 mmol/L dose (highest C/N ratio) inhibiting the activity of the methanogens; in contrast, the concentration of 2 mmol/L of ethanol and the resulting C/N ratio may be suitable for maintaining the activity of methanogens.

**Table 3.** The total carbon to total nitrogen (C/N) ratio driven by different ethanol in the IVCW.

| Parameters | Ethanol Concentrations | | | | | |
| --- | --- | --- | --- | --- | --- | --- |
| | 0 mmol/L (Blank Experiment) | 2 mmol/L | 4 mmol/L | 8 mmol/L | 16 mmol/L | 32 mmol/L |
| Inflow of TC (mg/L) | 0.6 | 0.7 | 1.3 | 1.9 | 3.8 | 6.8 |
| Outflow of TC (mg/L) | 0.6 | 0.3 | 9.3 | 5.8 | 2.6 | 8.0 |
| Inflow of TN (mg/L) | 4.2 | 2.5 | 2.4 | 2.0 | 2.2 | 3.5 |
| Outflow of TN (mg/L) | 1.6 | 1.8 | 2.5 | 1.7 | 1.7 | 1.0 |
| Inflow of C/N ratios (%) | 15.6 | 28.8 | 56.1 | 97.8 | 177.9 | 193.8 |
| Outflow of C/N ratios (%) | 36.9 | 192.1 | 374.1 | 339.2 | 152.2 | 801.1 |

### 3.3. $CH_4$-C Emission

The emission factors of methane included $CH_4$-C/TOC, $CH_4$-C/IC and $CH_4$-C/TC. The source of IC may come from the substrates, microbes, plant and atmosphere. Table 4 presents the share (%) of the $CH_4$ emission factors in the initial loading in the IVCW. These differences of three features can be used as the evaluation index for different circumstances. The $CH_4$-C/TOC, $CH_4$-C/IC and $CH_4$-C/TC decreased with increasing concentrations of 2, 4, 8, 16 and 32 mmol/L of ethanol concentrations. The average $CH_4$-C/TOC (%) value of 35% was driven by ethanol. Figures 6 and 7 present the $CH_4$-C/TOC features and $CH_4$-C/IC (inorganic carbon) features, respectively, under the different influent TOC in IVCWs ($p < 0.05$). These results can help clarify the dynamics of the ethanol assimilation. There was a significant multiple regression correction, as shown in Figure 7c ($R^2 = 0.903$) and Figure 7d ($R^2 = 0.944$).

**Table 4.** The share (%) of the $CH_4$ emission factors in the initial loading in the IVCW.

| Ethanol Concentrations | Average $CH_4$-C/TOC (%) | Average IC (g) | Average TC (g) | Average $CH_4$-C/IC (%) | Average $CH_4$-C/TC (%) |
| --- | --- | --- | --- | --- | --- |
| 0 mmol/L | 65 | 1.18 | 1.76 | 22 | 12 |
| 2 mmol/L | 66 | 1.33 | 2.02 | 44 | 26 |
| 4 mmol/L | 43 | 1.65 | 2.77 | 40 | 20 |
| 8 mmol/L | 24 | 2.5 | 4.94 | 32 | 9 |
| 16 mmol/L | 11 | 4.56 | 9.5 | 19 | 4 |
| 32 mmol/L | 3 | 9.68 | 19.91 | 16 | 1 |

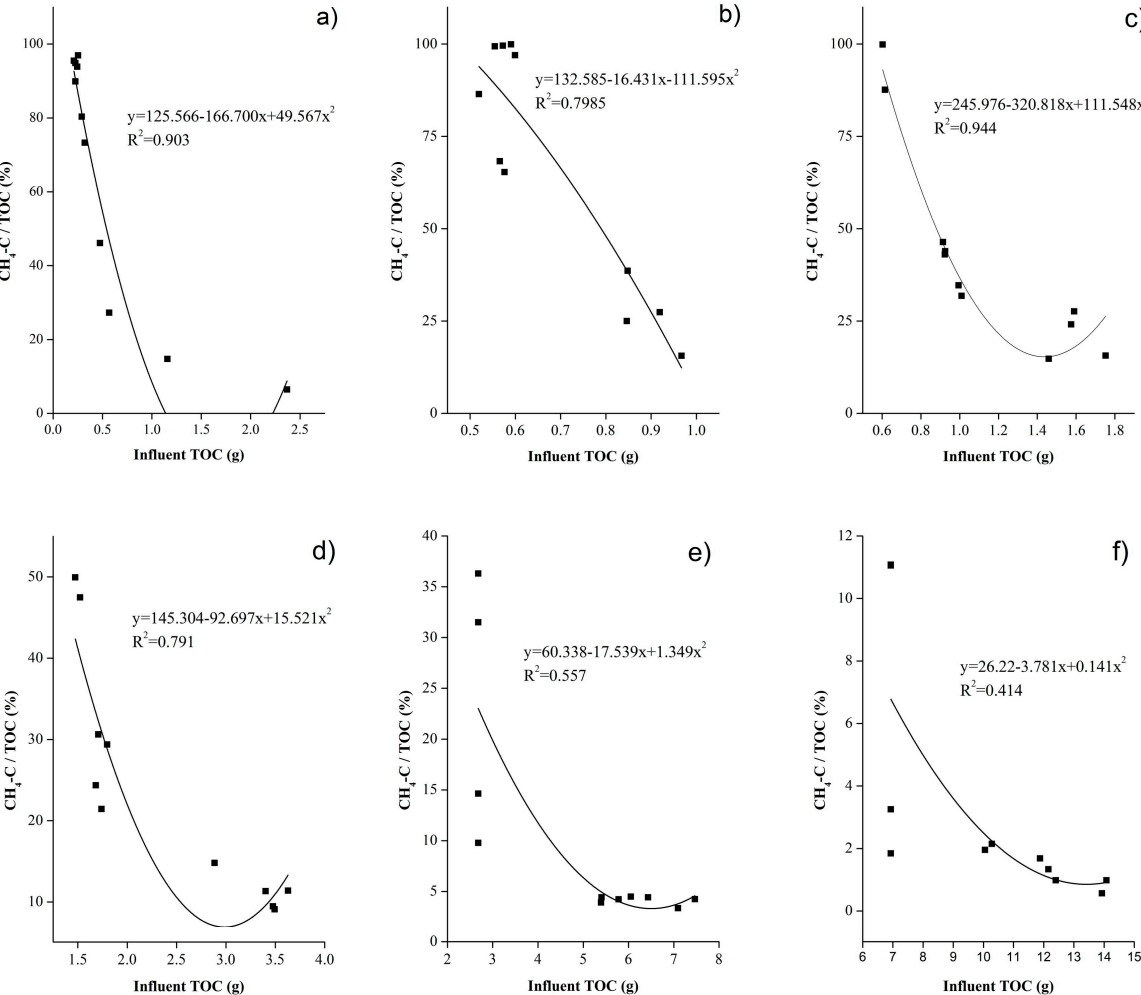

**Figure 6.** The relationship rate between the $CH_4$-C and influent total organic carbon (TOC). (**a**) 0 mmol/L ethanol added; (**b**) 2 mmol/L ethanol added; (**c**) 4 mmol/L ethanol added; (**d**) 8 mmol/L ethanol added; (**e**) 16 mmol/L ethanol added; (**f**) 32mmol/L ethanol added.

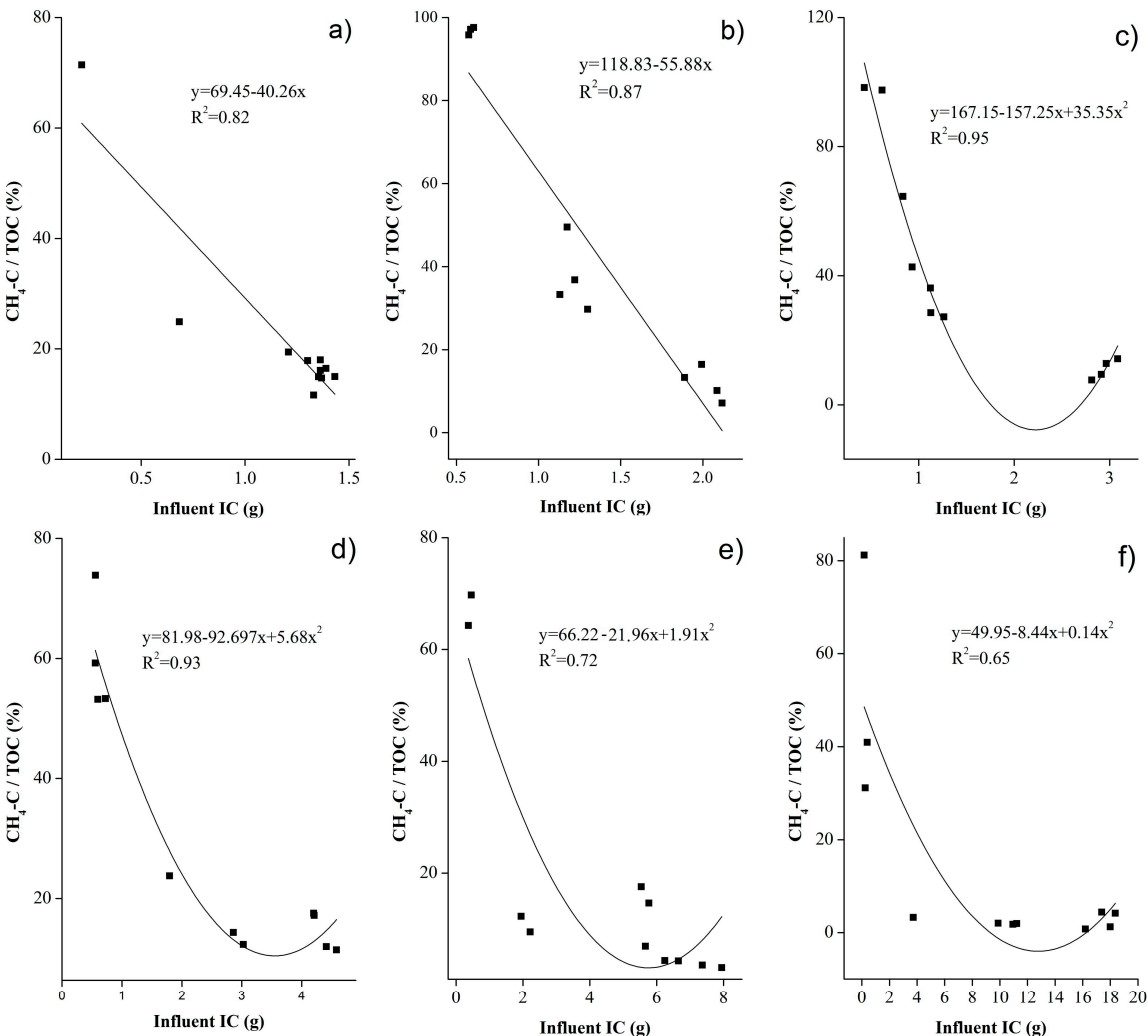

**Figure 7.** The relationship rate between the $CH_4$-C and influent inorganic carbon (IC). (**a**) 0 mmol/L ethanol added; (**b**) 2 mmol/L ethanol added; (**c**) 4 mmol/L ethanol added; (**d**) 8 mmol/L ethanol added; (**e**) 16 mmol/L ethanol added; (**f**) 32 mmol/L ethanol added.

## 4. Discussion

### 4.1. Methane Emission Flux Effects Driven by Ethanol and Water Quality

The results show that these concentrations (0–32 mmol/L) of ethanol negatively contributed to the methane emissions (Equation (3)), which were related to the carbon source. It has been reported that the added carbon was sufficiently labile to the support methanogenesis [61], but over a certain concentration of carbon source it was occurred that the inhibition of methanogenesis (for example, at some threshold level of 16–32 mmol/L of ethanol had occurred the inhibition of methanogenesis in this study). The carbon source and/or sink can also be affected by different types of CWs and environmental features. CWs can be both sources and sinks of carbon, depending on the environmental conditions. It has been found that only 15% of the carbon plants consumed in the *Phragmites* wetlands are released into the atmosphere, indicating that wetland ecosystems can restrain the elevated atmospheric $CO_2$ when using the wetland as a carbon sink. The temperature is the main factor affecting the oxygen content and aerenchyma of the plants; consequently, it affects the concentration of methane. This shows the importance of including the plant-derived material in the total carbon budget of constructed wetlands, which documents the importance of plants as a source of available carbon for microorganisms

in constructed wetlands that are not heavily loaded with wastewater. This carbon is further transformed into the gaseous form and increases the carbon emissions from the wetland [59].

A linear relationship between the temperature and methane emissions was discovered in this work ($y = 30.423x + 5779$ $R^2 = 0.86$), in some degree, consistent with previous studies. These studies found that different temperatures can significantly affect the methane emissions and noted that there are very few methane emissions below 15 °C [62–67]. The temperature and redox potential caused variations in the methane production by changing the activities of methanogens [63,68]. It was found that the highest $CH_4$ flux was obtained in the polyculture system, which measured the methane emissions ranging from $4.26 \pm 0.05$ to $9.16 \pm 0.08$ mg $CH_4$ day$^{-1}$ [69], and the results of this work were at a relatively high level of the methane emission from the IVCW compared to the author's research. In that study, it was shown that the average total amount of the $CH_4$ released from the non-vegetation control, polyculture and monoculture systems was 11.50, 92.01, and 47.61 mg $CH_4$ m$^{-2}$ h$^{-1}$, respectively [69]. Previous studies have observed the $CH_4$ fluxes from constructed wetlands ranging from $-2.12$ to 698.20 mg $CH_4$ m$^{-2}$ h$^{-1}$ [70,71].

According to Figure 3, the average of the $CH_4$ fluxes in the effluent tank was higher than that in the influent tank, and the root-water subsystem had a higher average than the stem-leaf subsystem did. The average value of the root-water subsystem may have been higher due to the effect of the root activity of plants and the stimulation of the methanogenic bacteria via the root exudation [72,73]. The ability to produce methane via the plant root's secretion of the acetic acid from fermentation is strong [74]. The methane emissions at the 32 mmol/L dose for each system were lowest compared with those of the other five concentrations, which might have been due to the lower pH (Figure 4a) in an acidic condition, ranging from 6.5 to 6.9. According to one study [75], significant pH effects were identified only when the temperatures were below 18.9 °C; however, the temperature of the 32 mmol/L of ethanol dose was in range of 8 to 10 °C (Figure 3c). The results were consistent with those of previous studies [76,77], where the pH was in a suitable range to sustain the denitrification process. There is one reason why the methane flux at the 32 mmol/L dose was the lowest: It is that the pH was beyond the optimum pH value of methanogenesis [74]. In addition, there might have been a temperature inhibition effect on the methane emissions. The temperature is an important factor that significantly affects the microbial community during the methane fermentation [78]. The difference in the pH is an important factor for the metabolism and biological diversity of methanogens. In addition, it is suitable for methanogens in neutral or alkaline environments [79].

*4.2. Methane Emission Impacted on C/N Ratio in IVCW*

The average 22.6% increase in the $CH_4$ emissions in this study is in accord with other research results, which found that the increased carbon loading (0–16 mmol/L ethanol) enhanced the $CH_4$ emissions. Therefore, an important conclusion drawn from this study is that adding ethanol to the IVCW could stimulate and increase the average $CH_4$ emissions from the IVCW by 23% compared to the control experiment (0 mmol/L). It has been found that the concentration of the carbon source affects the methane emissions by methanogens [80]. It was determined that the optimum C/N ratio is 5:1, at which point the VSSF CWs can achieve a relatively high biological nutrient removal efficiency and a low level of the GHG flux [58]. Another possible effect is that carbon compounds can stimulate methanogens [81,82]. Previous work has shown that the types and concentrations of the organic carbon may influence the methanotroph diversity [83,84], and it is agreed that it is necessary to keep a balanced composition for a balanced C/N ratio [85]. Other studies have concluded that the cumulative methane production increased from 0.17 L $CH_4$ to 2.03 L $CH_4$ in response to the treatment with a C/N ratio of 45. Since the maximum methane potential was achieved with a C/N ratio of 27.2:1 after optimization using the response surface methodology, it was noted that as the C/N ratio increased, the methane potential initially increased and then declined [86], which is a similar trend to that observed in this study. They suggested that the methane fluxes were lowest when the C/N ratio was seven, whereas the highest flux value was found when the C/N ratio was 26, which were low values compared with

those in this work. It can be explained that adding ethanol would drive up the C portion of the ratio. In addition, the C/N ratio of the external carbon source affects not only the methane emissions but also the removal of nitrogen. An increase in the C concentration results in a higher ratio, which stimulates the activity of the denitrifying bacteria and consequently changes the rate of removal of nitrogen because the carbon source is a controlling factor in the denitrification process.

### 4.3. Analysis of $CH_4$-C/TOC and $CH_4$-C/IC

There was a significant multiple regression correction between both the inflow TOC (in Figure 6) and IC (in Figure 7) loading values and the $CH_4$-C emission. The influent TOC were discussed and calculated based on the area, hydraulic load and inflow TOC concentration data from CWs, but the $CH_4$-C/TOC was not considered [38,87]. In this study, the $CH_4$-C/TOC and $CH_4$-C/IC had gradually decreased tendency in Figures 6 and 7 when the value of the influent TOC was increased during the experiment period. There was one polynomial trendline type of regression correction which was the secondary polynomial regression equation ($CH_4$-C/TOC = c + b × TOC + a × $TOC^2$, $R^2$ was 0.414 to 0.944) between the $CH_4$-C and TOC in Figure 6a–f. There was two trendline types of regression correction which was the secondary polynomial regression equation ($CH_4$-C/IC = c + b × IC + a × $IC^2$, $R^2$ was 0.65 to 0.95) between the $CH_4$-C and IC in Figure 7c–f, and another was the linear regression equation ($CH_4$-C/IC = b + a × IC, $R^2$ was 0.82 to 0.87) between $CH_4$-C and IC in Figure 7a,b. The median values of the $CH_4$-C/TOC values (percentage range and standard error (in parentheses)) of SF CWs, HSSF CWs, and VSSF CWs were 42.2 (18), 12 (4.15), and 1.17 (1.28), respectively [10]. The average values of the $CH_4$-C/TOC values in this study was higher than the report by the IPCC, and the main possible reason was that ethanol, as a kind of micro-molecular carbon chemical, can be much easily degraded by the microbe and plants in the IVCW than the macromolecular carbon chemical such as phenol, aromatic benzene, the hum-type substances, etc. [88]. As well as, the sewage wastewater composed of organic pollutants including fruit sugars, soluble proteins, drugs and pharmaceuticals are macromolecular weights than the value of ethanol molecular weight [89].

In order to calculate the carbon emission related carbon source, it is suitable to adapt the calculation ($CH_4$-C/carbon) by only the carbon contents from the methane and carbon, and to statistically compare with other researchers' greenhouse gas emission. Hence, the $CH_4$-C/TOC, $CH_4$-C/IC and $CH_4$-C/TC may be considered as the assessment factors of the anthropogenic methane emission when adding the carbon load in the IVCW. Tanner et al. [38] proposed estimated values of the inflow total organic carbon (TOC textsubscriptin) based on the area, hydraulic load and inflow TOC concentrations. An average of 35.3% of TOC is transformed into $CH_4$-C, which is due to the intensive accumulation of the organic matter [41]. Share (%) is the percentage of the GHG emission contribution to the atmosphere. The GHG emission factor contains every greenhouse gas such as the $CO_2$, $CH_4$, $N_2O$, fluorochloroparaffins, etc. A medium value of the share compared with those observed in previous studies is similar to the results of the previous research (35%) on dairy wastewater treatment [90]. The shares were 1.4% and 4% in the domestic wastewater treatment and 4% in the agricultural non-point source pollution treatment [91]. However, the share had a high value (79%) in a HSSF (Horizontal subsurface flow) constructed wetland [59], and the value reached 111.3%, which can be explained by the overloading and clogging of this kind of CW [41]. The non-linear relationship between the $CH_4$-C emissions and influent TOC may due to the increased concentration of ethanol limiting the activity of the methanogens or hydrolysis of the organic matter [42]. Only 10% of total carbon emissions were in the form of $CH_4$ [59]. The highest value for the emission factor (EF) of $CH_4$ (($CH_4$-C/$TOC_{in}$) × 100%) was found for SF CWs (median 18%), followed by HSSF CWs (3.8%) and VSSF CWs (1.28%). The mean $CH_4$-C/TOC values (percentage range and standard error (in parentheses)) of SF CWs, HSSF CWs, and VSSF CWs were 42 (20), 12 (6.9), and 1.17 (0.33), respectively [10]. The values of $CH_4$-C/TOC (%) in this study were higher than those in other studies [10,41]. The $CH_4$-C/TC can be determined from the total carbon contribution to the methane emissions.

The values of $CH_4$-C/IC (%) maybe used in some situations. For instance, the $CH_4$-C/IC (%) may establish contact with the distribution of the inorganic carbon in the runoff ocean carbon cycle [42]. For ICCW, the IC source may come from the metabolism and respiration by the microbes and plant, especially, the $CO_2$ dissolves into the water caused. The $CH_4$-C/IC (%) can find some relationship between the methane emission and inorganic carbon in the IVCW (in Figure 7). The $CH_4$-C/IC (%) is thus beneficial as one assessment factor for evaluating the transformation of greenhouse gases ($CO_2$, $CH_4$).

The $CH_4$-C/TOC is the main evaluation index in this study because most methane emissions come from the organic carbon transformation. In general, high concentrations of the organic wastewater, as in high TOC concentrations and loading, are commonly treated by CWs and sewage treatment systems. Therefore, the mean methane concentration may be determined by the value of the TOC during the wastewater treatment. Moreover, there is a significant correlation ($p = 0.02$) between the $CH_4$-C/TOC (%) and the ethanol concentration (($y_{share} = -5.73x^2 + 0.12x + 66.12$, $R^2 = 0.91$) (x: the ethanol concentration (mmol/L), $y_{share}$: the $CH_4$-C/TOC (%)). Therefore, with the $CH_4$-C emissions driven by ethanol, it will be very beneficial to use the $CH_4$-C/TOC (%) as an ecological indicator of $CH_4$ reduction effectively in wastewater treatment systems driven by carbon sources or carbon loading. Consequently, the values of $CH_4$-C/TOC (%) can be used for evaluating the wastewater treatment in the future.

### 4.4. Carbon Cycle in IVCW

The carbon cycle is also very important in IVCWs. The average influent carbon mass (TC, including dissolved compounds) when adding ethanol as the external carbon was 30.027 g, including the TOC of 26.17 g and IC of 3.857 g. the $CH_4$-C was 35.28% of the influent TOC mass. However, the average dissolved carbon mass (TC, TOC, IC) in treated water in the IVCW was higher than the influent carbon mass. This was one reason that it was the comprehensive function among the photosynthesis of *Cyperus alternifolius L.*, microbe metabolism and plant rhizosphere in the IVCW. There was a lack of $CO_2$ information that can be measured in future research.

The GWP in the IVCW was 0.58397 kg $CO_2$-eq GHG emissions in this study, only considering the $CH_4$ emissions. This result is significantly lower than treating the wastewater treatment by CW or WWTPs. It has been reported that the GWP was 3.11 kg $CO_2$-eq GHG emissions including $CH_4$ and $CO_2$, by the intermittent micro-aeration control of the methane emissions during the agricultural domestic wastewater treatment in the IVCW [92]. For the VFSS CWs system, it had mean GHG emissions of 3.18 kg $CO_2$-eq, and only half the amount of GHGs emitted by the WWTPs, in the meantime, the pulse pumping of VSSF CWs reed beds produces 2.7 kg $CO_2$-eq GHG emissions [93]. For the vertical flow constructed wetlands (VFCWs), it will have more efficient, lower energy and less environmental impact than HFCWs (horizontal flow constructed wetlands) due to its smaller footprint [94]. The total annual cost of this study was \$468/year (including construction fees, operation fees and maintainance fees, with the average treatment fee of \$3.75 per 1000 milliliter. Thus, there is probably a much cheaper solution to add an additional C source (e.g., biochar) or other chemical elements (e.g., $SO_4^{2-}$) to reduce the $CH_4$, compared with this study. When the carbon source contained ethanol in the IVCW, the concentration of ethanol was over 32 mmol/L for the $CH_4$ emission controls of the application in a full-scale IVCW. In the future, more details of the carbon cycles in the IVCW should be determined.

### 5. Conclusions

A gradient of ethanol as a carbon source had a significant impact on the methane emissions in the IVCW in this study. Compared with the control experiment (0 mmol/L ethanol), the methane flux stimulated by ethanol increased under 16 mmol/L of ethanol concentration, then it was an inhibition of the metahnogenesis in the IVCW with above 16 mmol/L of ethanol. The highest average carbon usage rate and removal rate of COD in the IVCW from the added ethanol were observed when 16 mmol/L of ethanol was added. The IVCW water purification efficiency and decreasing greenhouses gas emissions are two important goals that must be met together in the constructed wetland. The $CH_4$-C/TOC can be

considered an ecological indicator of the anthropogenic methanogenesis from treated wetlands driven by carbon sources. Since the methane emissions can be calculated using the value of the $CH_4$-C/TOC, this value can consequently be used to evaluate the wastewater treatment effect in CWs and sewage treatment plants and is suitable for use in the life cycle assessment.

**Author Contributions:** Conceptualization, X.L. and H.L.; Data curation, X.L.; Formal analysis, K.Z., L.F., L.Z., S.F., L.H. and M.J.; Funding acquisition, H.L.; Investigation, M.L., L.Y., G.H. and L.H.; Methodology, H.L.; Experiment, K.Z.; Project administration, H.L.; Supervision, H.L.; Visualization, B.H.; Writing—original draft, J.W.; Writing—review & editing, X.F. and B.C.A.

**Funding:** The work was supported by the National Natural Science Foundation of China (No. 51278318, 51808363), Key Lab of Hydraulics and Mountain River Engineering (No. SKHL1716), Key Development Program from the Science and Technology Department of Sichuan Province (No.2013SZ0103, 18ZDYF3209), and Chengdu Science & Technology Bureau (2015-HM01-00325-SF).

**Conflicts of Interest:** The authors declare no conflict of interest.

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
