# Peer review of "Methane Emissions Driven by Adding a Gradient of Ethanol as Carbon Source in Integrated Vertical-Flow Constructed Wetlands"

_water, doi:10.3390/w11051086_

Round 1

Reviewer 1 Report

The research results presented by the authors in ““Methane Emissions Driven by Adding a Gradient of Ethanol as a Carbon Source in Integrated Vertical-Flow Constructed Wetlands” are interesting and may be used in further experiments.

At the moment I feel  there is a lack of information about the necessity of your experiment. In my opinion you should give some more information about external carbon sources in constructed wetland’s treatment. In addition please justify your use of the IV CW system. It is not a very popular system for water and sewage treatment.

Abstract

Line 39:

It is difficult to understand what is meant by “control experiment”.

Line 36:

Just explain what does “treated wetlands” mean.

Introduction

Line 52:

Please do not use terms such as special wastewater treatment. What is “better sewage treatment effect” ?

Line 61:

It is true that we can find the problem with methane emission from municipal and industrial WWTPs utilizing biological treatment but try to justify the problem of methane emission from wetlands. It is not possible to avoid emission from natural wetlands. How big (in total) is the emission from constructed wetlands ???

Line 101:

 In my opinion there is no need to supply constructed wetlands treating typical sewage (household) or used as a final treatment after biological methods. Maybe in some cases (not typical sewage) we can supply CW with external carbon. Applying external carbon in constructed wetland technology seems to be an imaginary problem.

Line 129:

Also you can write about emissions from sewage sludge stabilization in municipal and industrial WWTPs.

Line 15:4

In my opinion it is insufficient to use “little knowledge” and create your experiment. 

Line 156:

Please explain what “water purification” means.  What kind of water??

Material and methods

Line 176:

What was the reason for applying 5 layers ?

Line 202:

You wrote about flow. What was the hydraulic load in m3/m2d (m/d)

Line 223

What was the reason for a 120 h duration time ?

Results and Discussion

Figure 2

Is it possible to prepare figures in better resolution ? It will simplify the understanding and analyzing of the presented results.

Line 336

Not clear, correct your language please.

I also have  some problems with figure 8. Maybe you can simplify it by removing some information.

Conclusions

Line 521

The sentence is not clear “evaluate wastewater” ???

References
I think that you should decrease the quantity of references. It will not diminish the research value of your paper. This is only my suggestion. Also try to correct your English.

Author Response

Response to Reviewer 2 Comments

Thank reviewers and editors very much for reviewing our manuscript (water-473854), we have revised and answered carefully our manuscript based on the opinions and suggestions from reviewers and editors. Response to reviewers and editor are bellows (relative text including the Grammar and writing were in red fonts in our revised manuscript):

We have substantially revised our manuscript after reading the comments provided by the editors and two reviewers (Reviewer 2). Please see following response:

Reviewer 2

Point 1:

Abstract

Line 39:

It is difficult to understand what is meant by “control experiment”.

Response 1: Thank for very much.

Control experiment usually be considered as the comparison in order to find the difference from other experiments.

Control experiment also be considered as the blank experiment that the concentration is zero.

We had firstly expressed in Line 25 (0 mmol/L ethanol).

Point 2: Line 36:

Just explain what does “treated wetlands” mean.

Response 2: Thank for very much.

We had revised it as the “treatment wetland”.

Point 3:

Introduction

Line 52:

Please do not use terms such as special wastewater treatment. What is “better sewage treatment effect” ?

Response 3: Thank for very much.

We had revised “better sewage treatment effect” as the “nice wastewater treatment effect”

Point 4:

Line 61:

It is true that we can find the problem with methane emission from municipal and industrial WWTPs utilizing biological treatment but try to justify the problem of methane emission from wetlands. It is not possible to avoid emission from natural wetlands. How big (in total) is the emission from constructed wetlands ???

Response 4: Thank for very much.

We added the methane emission from nature wetland. Climate-driven fluctuations of CH4 emissions from natural wetlands (177 to 284 ×1012 g (CH4) yr–1 (IPCC, 2013).

Anthropogenic emissions account for 50 to 65% of total emissions of global methane.

Please see the second paragraph of section of 1 (1. Introduction).

Point 5:

Line 101:

In my opinion there is no need to supply constructed wetlands treating typical sewage (household) or used as a final treatment after biological methods. Maybe in some cases (not typical sewage) we can supply CW with external carbon. Applying external carbon in constructed wetland technology seems to be an imaginary problem.

Response 5: Thank for very much.

As reviewer 2 mentioned, not all CW need to add external carbon source. This paper just made a try to investigate the driven effect on methane emission from an IVCW by adding ethanol as external carbon. It is meaningful to debate the possible impact on methane emission from CW when add carbon source.

Point 6:

Line 129:

Also you can write about emissions from sewage sludge stabilization in municipal and industrial WWTPs.

Response 6: Thank for very much.

We added relevant contents.

Methane emission were mainly from sewage sludge stabilization in municipal and industrial WWTPs.

Please see the fourth paragraph of section of 1 (1. Introduction).

Point 7:

Line 154

In my opinion it is insufficient to use “little knowledge” and create your experiment.

Response 7: Thank for very much.

We revised “little knowledge” as the “no any knowledge”.

Point 8:

Line 156:

Please explain what “water purification” means. What kind of water??

Response 8: Thank for very much.

We revised the “water purification” as the “wastewater purification”

Point 9:

Material and methods

Line 176:

What was the reason for applying 5 layers ?

Response 1: Thank for very much.

We selected five layers as structures of IVCW, based on previous research, from the reference of Luo et al. (2015).

We added this reference here.

Point 10:

Line 202:

You wrote about flow. What was the hydraulic load in m3/m2d (m/d)

Response 10: Thank for very much.

We revised the “influent flow velocity” as the “influent velocity of ethanol solution ”.

The hydraulic load unit is the m3/m2/d.

Point 11:

Line 223

What was the reason for a 120 h duration time ?

Response 11: Thank for very much.

Based on the reference of Deng et al., (2011) about the metabolic stability of methanogensis after 96 hours, we think how it is states if time once more one day (i.e. 120 hours ). It’s somehow meaning to select 120 hours as experiment time limit.

Point 12:

Results and Discussion

Figure 2

Is it possible to prepare figures in better resolution ? It will simplify the understanding and analyzing of the presented results.

Response 12: Thank for very much.

We think that using Fig. 2 can clearly express the data change in ethanol of 0, 2, 4, 8, 16 and 32 mmol/L with sampling times, respectively, in order to find methane flux change.

Point 13:

Line 336

Not clear, correct your language please.

I also have some problems with figure 8. Maybe you can simplify it by removing some information.

Response 13: Thank for very much.

We had deleted Fig. 8 in this revised edition. Because some information was not clear, possibly misunderstanding readers.

Point 14:

Conclusions

Line 521

The sentence is not clear “evaluate wastewater” ???

Response 14: Thank for very much.

We revised the“evaluate wastewater”as the “evaluate wastewater treatment effect”.

Point 15:

References
I think that you should decrease the quantity of references. It will not diminish the research value of your paper. This is only my suggestion. Also try to correct your English.

Response 15: Thank for very much.

We had deleted three references, which not diminish the research value. These reference are as follows:

Oliver, J.E., 2013. Intergovernmental Panel in Climate Change (IPCC). Encyclopedia of Energy Natural Resource & Environmental Economics 26, 48-56.

Zhu, G., Jetten, M.S.M., Kuschk, P., Ettwig, K.F. and Yin, C. 2010. Potential roles of anaerobic ammonium and methane oxidation in the nitrogen cycle of wetland ecosystems. Applied Microbiology & Biotechnology 86(4), 1043-1055.

Gómez, M.A., Hontoria, E., González-López, J., 2002. Effect of dissolved oxygen concentration on nitrate removal from groundwater using a denitrifying submerged filter. J. Hazard. Mater. 90, 267-278.

We also revised some little wrong in reference. Please see the red fonts.

We had correct our English language carefully. We had uploaded our English certificate.

Reviewer 2 Report

This is interesting paper but I have some questions:

1) Nice experiments but how is it applicable in full-scale conditions? How much ethanol you need to add to the full-scale CW to reduce methane emissions? What would be the annual cost, if you would add it to the CW treating a wastewater from 1000 PE (for example?). I would like to see some feasibility discussion because there are probably much cheaper solution to add additional C source (e.g. biochar) or other elements (SO4) to reduce CH4.

2) Figure 6 and 7 includes total of 12 sub-figures which are not discussed at all. If you add so many figure with different trends, they must be discussed. 

3) Most of the figures have different text sizes. Please unify these.

4) Figure 4 - would like to see is there statistical differences between treatments. Also what does bars show - min, max, median, mean, 25%-75%, etc? Specify? 

5) I don't like figure 8 because it is over-simplified and can be misleading. You don't have any information about the activity of methanotrophs (or its potential) and there is no information about C loss via effluent water. 

6) Conclusions of the study are very tiny. Would like more concrete conclusion showing what we actually learnt from this experiment/study.

Author Response

Response to Reviewer 1 Comments

Thank reviewers and editors very much for reviewing our manuscript (water-473854), we have revised and answered carefully our manuscript based on the opinions and suggestions from reviewers and editors. Response to reviewers and editor are bellows (relative text including the Grammar and writing were in red fonts in our revised manuscript):

We have substantially revised our manuscript after reading the comments provided by the editors and two reviewers (Reviewer 1). Please see following response:

Reviewer 1

Point 1: Nice experiments but how is it applicable in full-scale conditions? How much ethanol you need to add to the full-scale CW to reduce methane emissions? What would be the annual cost, if you would add it to the CW treating a wastewater from 1000 PE (for example?). I would like to see some feasibility discussion because there are probably much cheaper solution to add additional C source (e.g. biochar) or other elements (SO4) to reduce CH4.

Response 1: Thank for very much.

When application, carbon source of ethanol as liquid solution, can be determined and induced into a full-scale CW. Based on our investigation, ethanol can stimulated methane emission from IVCW, therefore, the concentration of ethanol was over 32 mmol/L for the CH4 emission control if application in a full-scale IVCW.

Please see before the section of Conclusion.

The annual cost was $468/year, with the average treatment fee of $3.75 per 1000 PE. Thus, there are probably much cheaper solution to add additional C source (e.g. biochar) or other chemical elements (e.g. SO42-) to reduce CH4, compared with this study.

Please see before the section of Conclusion.

Point 2: Figure 6 and 7 includes total of 12 sub-figures which are not discussed at all. If you add so many figure with different trends, they must be discussed.

Response 2: Thank you very much.

There were some discussion in the section of 4.3 (4.3Analysis of CH4-C/TOC and CH4-C/IC), compared with other research results.

Point 3: Most of the figures have different text sizes. Please unify these.

Response 3: Thank for very much.

When we draw our figures, we had unified same sizes of figure’s text. There was only one reason that some Figures were zoomed differently. We had revised these Figures.

Point 4: Figure 4 - would like to see is there statistical differences between treatments. Also what does bars show - min, max, median, mean, 25%-75%, etc? Specify?

Response 4: Thank for very much.

We had added relevant statistical data from Figure 4.

Please see the fifth paragraph of the section of 3.1 ( 3.1Methane emissions and effects on water quality driven by ethanol addition in the IVCW).

Point 5: I don't like figure 8 because it is over-simplified and can be misleading. You don't have any information about the activity of methanotrophs (or its potential) and there is no information about C loss via effluent water.

Response 5: Thank for very much.

We had deleted Fig. 8 in this revised edition.

Point 6: Conclusions of the study are very tiny. Would like more concrete conclusion showing what we actually learnt from this experiment/study.

Response 6: Thank for very much.

There were two important findings in this study. One was that less 16 mmol/L ethanol as carbon source had stimulated methane emission in IVCW. Another was the CH4-C/TOC can be considered an ecological indicator of anthropogenic methanogenesis from treated wetlands driven by carbon sources. We had expressed it in our conclusion.

Round 2

Reviewer 2 Report

Authors have made minor corrections but most things are still missing or not well explained. For example, Figure 6 and 7 includes total of 12 sub-figures which are not discussed at all. On your response you noted "There were some discussion in the section of 4.3". I agree, you have cited fig 6 once and fig 7 twice. However, there is absolutely no discussion what does all the sub-figures stand for (a-f in  Fig. 6 anf Fig. 7). All you are saying is that there was multiple significant correlations. But what do these all correlations mean? There are different trends, so what does these trends show us? Without any explanations these are just figures where points are connected with different functions but without any explanation. All of these subfigures must be discussed. Secondly, I would like to see statistical differences in Figure 4, so it would stand alone and readers don't need to search the text to find which of the parallels were different. And again, what does bars show - min, max, median, mean, 25%-75%, etc? Specify?  

Why has 16 mmol ethanol highest removal rate? Discussion is very brief and I would like to see more detailed insight how different ethanol concentrations affect microbes and why?

Author Response

Response to Reviewer 2 Comments

Thank reviewers and editors very much for reviewing our manuscript (water-473854) after first finished revision, we have revised and answered carefully our manuscript based on the opinions and suggestions from reviewers and editors. Response to reviewers and editor are bellows (relative text including the Grammar and writing were in red fonts in our revised manuscript):

We have substantially revised our manuscript after reading the comments provided by the editors and two reviewers (Reviewer 2). Please see following response:

Reviewer 2

Point 1:

Authors have made minor corrections but most things are still missing or not well explained. For example, Figure 6 and 7 includes total of 12 sub-figures which are not discussed at all. On your response you noted "There were some discussion in the section of 4.3". I agree, you have cited fig 6 once and fig 7 twice. However, there is absolutely no discussion what does all the sub-figures stand for (a-f in Fig. 6 and Fig. 7). All you are saying is that there was multiple significant correlations. But what do these all correlations mean? There are different trends, so what does these trends show us? Without any explanations these are just figures where points are connected with different functions but without any explanation. All of these subfigures must be discussed.

Response 1: Thank for very much.

We had discussed for Fig. 6 and Fig.7 in the first paragraph in the Section of 4.3. Analysis of CH4-C/TOC and CH4-C/IC, as follows:

 There was a significant multiple regression correction between both the inflow TOC (in Fig. 6) and IC (in Fig. 7) loading values and CH4-C emission. The influent TOC were discussed and calculated based on area, hydraulic load and inflow TOC concentration data from CWs, but the CH4-C/TOC was not considered (Tanner et al., 1997; Gr Nlund et al., 2006). In this study, CH4-C/TOC and CH4-C/IC had gradually decreased tendency in Fig. 6 and Fig. 7 when the value of influent TOC was increased during experiment period. There was one polynomial trendline type of regression correction which it was the secondary polynomial regression equation (CH4-C/TOC = c + b×TOC + a×TOC2, R2 was 0.414 to 0.944) between CH4-C and TOC in Fig. 6a) to 6f). There was two trendline types of regression correction which it was the secondary polynomial regression equation (CH4-C/IC = c + b×IC + a×IC2, R2 was 0.65 to 0.95) between CH4-C and IC in Fig. 7c) to 7f), and another was the linear regression equation (CH4-C/IC = b + a×IC, R2 was 0.82 to 0.87 between CH4-C and IC in Fig. 7a) to 7b)). The median values of CH4-C/TOC values [percentage range and standard error (in parentheses)] of SF CWs, HSSF CWs, and VSSF CWs were 42.2 (18), 12 (4.15), and 1.17 (1.28), respectively (IPCC, 2013). The average values of CH4-C/TOC values in this study was higher than the report by IPCC, and the possible mainly reason was that ethanol, as a kind of micro-molecular carbon chemical, can be much easily degraded by microbe and plants in IVCW than macromolecular carbon chemical such as phenol, aromatic benzene, the hum-type substances, etc (Rodríguez-Murillo et al., 2011). As well as, sewage wastewater composed of organic pollutants including fruit sugars, soluble proteins, drugs and pharmaceuticals are macromolecular weights than value of ethanol molecular weight (WWAP, 2017).

And we supplied three reference in the reference area, as follows:

Grønlund, A., T.E. Sveistrup, A.K. Søvik, D.P. Rasse and B. Kløve, 2006. Degradation of cultivated peat soils in northern norway based on field scale CO2, N2O and CH4 emission measurements. Arch Agron Soil Sci, 52: 149-159.

WWAP (United Nations World Water Assessment Programme), 2017. The United Nations World Water Development Report 2017. Wastewater: The Untapped Resource, Paris.

Rodríguez-Murillo, J.C., G. Almendros and H. Knicker, 2011. Wetland soil organic matter composition in a Mediterranean semiarid wetland (Las Tablas de Daimiel, Central Spain): Insight into different carbon sequestration pathways. Org Geochem, 42: 762-773.

Point 2: Line 36:

Secondly, I would like to see statistical differences in Figure 4, so it would stand alone and readers don't need to search the text to find which of the parallels were different. And again, what does bars show - min, max, median, mean, 25%-75%, etc? Specify?    

Response 2: Thank for very much.

The min, median, mean and max value of the removal rate of COD was 41%, 48.3%, 51% and 78%, respectively, in Fig. 4a. The min, median, mean and max value of the removal rate of ammonia nitrogen was 10.2%, 58%, 57%, and 93%, respectively, in Fig. 4b.

   In the Fig. 4, the bars was modified to show the value of 10%, 25%, 50% and 75% by box plot. Please see the revised Fig. 4.

Fig.4 Removal rate of COD and NH3-N

Point 3:

Why has 16 mmol ethanol highest removal rate? Discussion is very brief and I would like to see more detailed insight how different ethanol concentrations affect microbes and why? Response 3: Thank for very much.

The highest removal rate of COD (16 mmol/L dose) was 78% at 18 hours, which the average pH value of 16 mmol/L ethanol was highest and possibly accelerated COD degradation.

We hadn’t investigated the microbe and plant change in studied IVCW (in the section of 2. Material and methods). We will monitor microbe in future. Therefore, we made a conclusion as follows:

The removal rate of ammonia nitrogen increased after adding 2 mmol/L, 4 mmol/L, 8 mmol/L and 16 mmol/L of ethanol compared with 0 mmol/L of ethanol, which shows that a carbon source less 16 mmol/L ethanol could effectively stimulate biodegradation in this CWs.

Please see the line 318 to 327.

Round 3

Reviewer 2 Report

After second revision authors have provided sufficient amount of new and important information and therefore recommend to accept the paper.